# An Auxiliary Task for Learning Nuclei Segmentation in 3D Microscopy Images

**Peter Hirsch**                                        PETER.HIRSCH@MDC-BERLIN.DE
**Dagmar Kainmueller**                      DAGMAR.KAINMUELLER@MDC-BERLIN.DE
*Berlin Institute of Health /*
*Max-Delbrueck-Center for Molecular Medicine in the Helmholtz Association*

## Abstract

Segmentation of cell nuclei in microscopy images is a prevalent necessity in cell biology. Especially for three-dimensional datasets, manual segmentation is prohibitively time-consuming, motivating the need for automated methods. Learning-based methods trained on pixel-wise ground-truth segmentations have been shown to yield state-of-the-art results on 2d benchmark image data of nuclei, yet a respective benchmark is missing for 3d image data. In this work, we perform a comparative evaluation of nuclei segmentation algorithms on a database of manually segmented 3d light microscopy volumes. We propose a novel learning strategy that boosts segmentation accuracy by means of a simple auxiliary task, thereby robustly outperforming each of our baselines. Furthermore, we show that one of our baselines, the popular three-label model, when trained with our proposed auxiliary task, outperforms the recent STARDIST-3D.

As an additional, practical contribution, we benchmark nuclei segmentation against nuclei *detection*, i.e. the task of merely pinpointing individual nuclei without generating respective pixel-accurate segmentations. For learning nuclei detection, large 3d training datasets of manually annotated nuclei center points are available. However, the impact on detection accuracy caused by training on such sparse ground truth as opposed to dense pixel-wise ground truth has not yet been quantified. To this end, we compare nuclei detection accuracy yielded by training on dense vs. sparse ground truth. Our results suggest that training on sparse ground truth yields competitive nuclei detection rates.

**Keywords:** machine learning, image analysis, instance segmentation, instance detection, nuclei segmentation, auxiliary training task

## 1. Introduction

Locating and segmenting cell nuclei is often the first step in many analyses of biological processes on the cell level and their reaction to potential new medical treatments. Manual segmentation of nuclei in three-dimensional microscopy images is tedious. Hence 3d benchmark image data with pixel-wise ground truth labelings are barely available.[1] Instead, 3d benchmark image data predominantly come with sparse ground truth annotations in the form of nuclei center point locations (see e.g. (Ulman et al., 2017)). Consequently, methods for learning 3d nuclei segmentation based on pixel-wise, *dense* ground truth labelings remain understudied to date. Related work is focused on learning nuclei *detection* based on

---

1. This is opposed to large fully annotated 2d benchmark datasets, see e.g. image set BBBC038v1 available from the Broad Bioimage Benchmark Collection (Ljosa et al., 2012)

the given *sparse* ground truth (Ulman et al., 2017; Höfener et al., 2018), or for 3d segmentation on leveraging very small sets of 2d slices with dense labelings (Çiçek et al., 2016) or bounding boxes together with a low number of pixel-wise annotated instances (Zhao et al., 2018).

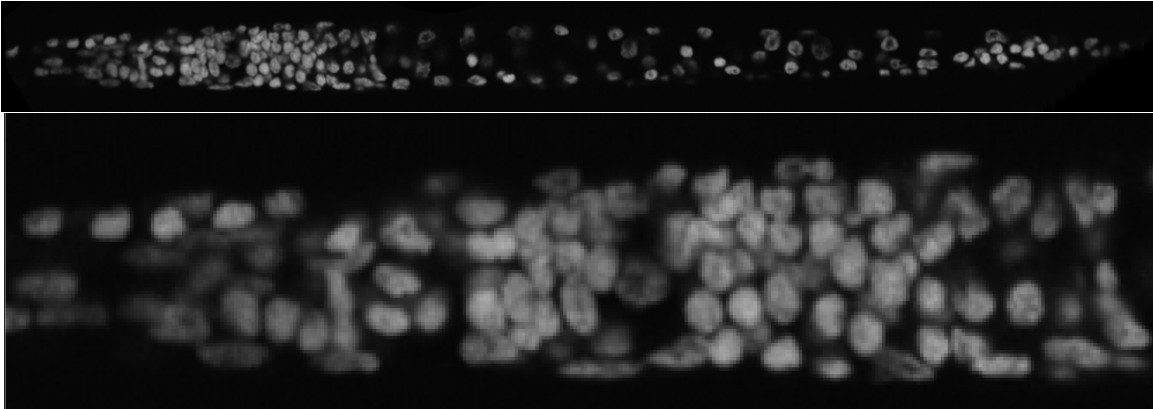

Figure 1: Top: Exemplary slice of an image in the dataset. Densely packed nuclei in the nervous system of the *C. elegans* L1 larva (towards the left) are particularly hard to separate, in some cases even by eye. Bottom: Close-up on said nervous system.

In this work, we employ a densely annotated dataset of 28 three-dimensional microscopy images described in (Long et al., 2009), each containing hundreds of nuclei, to benchmark 3d nuclei segmentation methods trained on dense pixel-wise ground truth.[2] Figure 1 shows an exemplary image slice. We focus on methods that yield state-of-the-art results for nuclei segmentation in 2d data, and have been established for generic 3d segmentation applications. In particular, we focus on U-Net based architectures (Ronneberger et al., 2015; Çiçek et al., 2016) for pixel-wise training and prediction. Furthermore, we compare our results to (Weigert et al., 2019) who recently reported results on the same dataset.

Nuclei segmentation is an instance segmentation problem, where the challenge is to separate object instances that appear in dense clusters. Existing U-Net based instance segmentation approaches that are directly applicable to 3d nuclei segmentation rely on classification of boundary pixels (Chen et al., 2016; Guerrero-Pena et al., 2018; Caicedo et al., 2019; Falk et al., 2019), classification of boundary edges between pixels (Funke et al., 2018), or regression of signed distances to boundary pixels (Heinrich et al., 2018). These methods have in common the property that small variations in the predictions at individual pixels can lead to drastic changes in the resulting instance segmentation; e.g., misclassifying a single boundary pixel as foreground can lead to a false merge of two nuclei.

In this work, we propose a novel auxiliary task that is designed to address this issue during training. In particular, we propose to train for predicting vectors pointing to the center of mass of the containing nucleus as an auxiliary task in addition to training for predicting the usual output variables. Note that training such offset vectors in itself is

---

2. We thank the authors of (Long et al., 2009) for providing the data.

not new, (Xie et al., 2015) locate the points with the highest number of incoming vectors for nuclei detection and (Li et al., 2018) use them not as an auxiliary task but in a split network branch, the output of which is used for additional post-processing. In contrast, we do not use the vectors for inference, only for training. We show in an evaluation on 28 3d microscopy images that training for the proposed auxiliary task boosts the accuracy of each underlying baseline model. Furthermore, our baseline three-label model (Caicedo et al., 2019), when trained with the proposed auxiliary task, outperforms the proposal-based approach StarDist-3D (Weigert et al., 2019) in terms of average AP[3] .

In addition to our methodological contribution, our work also provides a practical contribution. Due to the lack of densely annotated benchmark 3d nuclei images, to our knowledge, the impact of training on dense vs. sparse ground truth for 3d nuclei *detection* has not yet been quantified. In this work we fill this gap by contributing a quantitative comparison of densely- vs. sparsely trained 3d nuclei detection. In particular, we compare a state-of-the-art sparsely trained method that regresses Gaussian blobs of fixed radius around center points (Höfener et al., 2018) to the densely trained U-Net based architectures considered in the course of our methodological contribution. Our evaluation shows that training on sparse annotations in the form of nuclei center points yields competitive detection rates on par with baseline segmentation methods, but is outperformed by models trained with our auxiliary task.

In summary, we contribute (1) a novel auxiliary training task that boosts the accuracy of underlying baseline models for 3d nuclei segmentation, and (2) a quantitative comparison of densely- vs. sparsely trained methods for nuclei detection. Our code for all compared methods is available on https://github.com/Kainmueller-Lab/aux_cpv_loss/tree/arxiv.

## 2. Method

Methods for pixel- or edge-wise boundary prediction share the property that they rely on a small number of training pixels for learning to distinguish densely packed clusters of objects from larger, solitary objects, as illustrated in Figure 2B. Existing methods approach this issue by up-weighting the boundary class with a fixed factor (Caicedo et al., 2019), by adaptive up-weighting of boundary misclassifications that cause large segmentation errors (Funke et al., 2018), or by phrasing the boundary prediction task as a regression problem on the (signed) Euclidean distance transform to the boundary pixels (filtered by some smooth capping function) (Bai and Urtasun, 2016; Heinrich et al., 2018).

Instead, we propose a simple auxiliary training task that does not require any explicit, optimized weighting scheme nor any architectural changes and is nevertheless able to focus the training on reducing misclassifications that induce large segmentation errors. Specifically, we propose to regress vectors that point from each foreground pixel to the center of mass of the respective nucleus (see Figure 2C). We employ a sum of squared differences loss on these center point vectors, and add that to the main pixel classification or regression loss. Thus, in case a cluster of objects is mistaken for a single object or vice-versa, *all* foreground pixels of the involved object(s) contribute to form a large loss, instead of just a small subset of (up-weighted) boundary pixels as in (Caicedo et al., 2019; Funke et al., 2018), or pixels close to these as in (Heinrich et al., 2018).

---

3. avAP: mean average precision, averaged over a range of IoU thresholds

We found our auxiliary task to be most beneficial when training for *absolute vectors to center points* instead of unit vectors pointing towards center points. Note that unit vectors pointing away from object boundaries have been considered with a similar motivation for instance segmentation in 2d natural images (Bai and Urtasun, 2016), yet not as auxiliary task but to pre-train part of a model that regresses the distance transform to boundary pixels. Training for absolute vectors is facilitated in our case of nuclei segmentation because of the relatively small size and roughly ellipsoidal shape of nuclei.

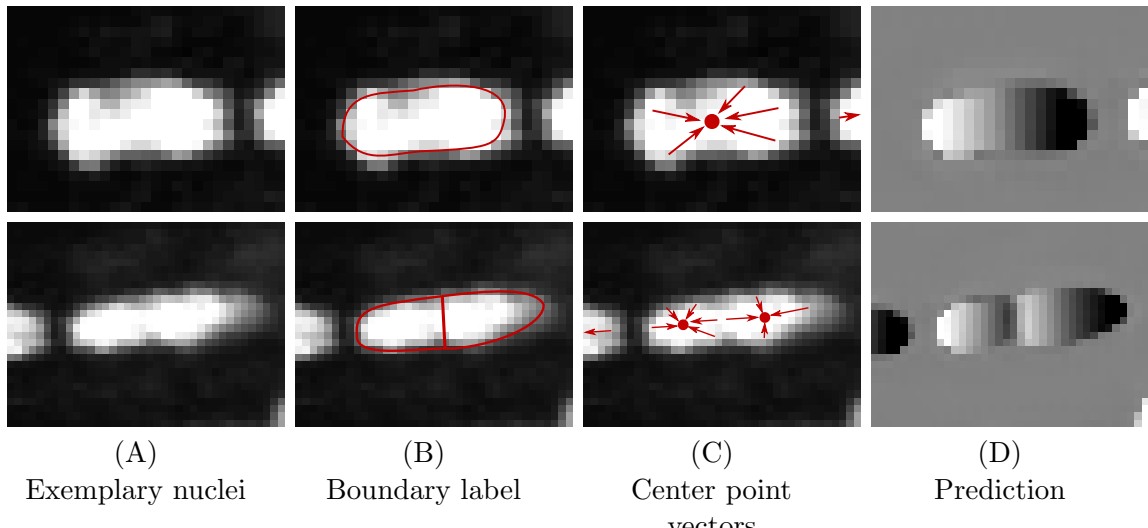

|           (A)            |           (B)           |           (C)            |          (D)           |
| Exemplary nuclei | Boundary label | Center point vectors | Prediction |

Figure 2: Illustration of the benefit of center point vectors as auxiliary training task. 1st row: Image detail focused on a single nucleus. 2nd row: Two densely packed nuclei, hard to distinguish by eye. (A) Raw image details. (B) Sketch of boundary pixels. (C) Sketch of a few exemplary center point vectors. While boundary labels distinguish between the presence of two vs. one nucleus at only the few pixels where two nuclei touch, center point vectors exhibit large differences at all foreground pixels. (D) Center point vectors per pixel predicted as auxiliary variables by our proposed model (3d, only the x-component is shown here).

Altogether, the proposed auxiliary task is easily integrated into standard end-to-end trainable models, with the only required architectural change being three additional output channels (one for each component of the 3d center point vectors). The one other training component that is affected by integrating our auxiliary task is elastic augmentation, where we generate training center point vectors on the fly. This can be done efficiently so that training performance is not considerably affected (For more details see Appendix B).

## 3. Experiments

We evaluate our methods and baselines on a set of 28 3d confocal microscopy images of wild type *C. elegans* at the L1 larval stage, as described in (Long et al., 2009). The nuclei of all cells were stained with DAPI, and all nuclei that can be distinguished by eye were segmented

manually. Each 3d image captures a single *C. elegans* larva. Due to the stereotypical nature of *C. elegans*, this amounts to about 558 nuclei per image. All images have a near-isotropic voxel size of $0.116 \times 0.116 \times 0.122 \mu m^3$ and an average size of $140 \times 140 \times 1100$ pixel. To be able to compare our results to (Weigert et al., 2019) we use the same data split. Accordingly, the dataset consists of a training set of 18 images, a validation set of 3 images, and a test set of 7 images. To verify our results further, we additionally perform 4-fold cross-validation (see Appendix C and Table 3) For the segmentation task we compare three baseline models, each trained with and without our proposed auxiliary task:

1. A model with a single scalar output regressing the signed Euclidean distance transform to the boundary pixels filtered by a tanh function, with sum of squared differences loss(*sdt*, see (Heinrich et al., 2018))

2. A 3-label model trained to classify background, foreground, and boundary pixels with a softmax cross entropy loss (*3 label*, see (Caicedo et al., 2019))

3. An edge affinity model for pairwise, direct neighbor affinities with binary cross entropy loss (*affinities*, see (Fowlkes et al., 2003; Funke et al., 2018)).

We denote training with our auxiliary task of regressing nuclei center point vectors as *+cpv*. We use a 3 layer 3d U-Net backbone architecture in all experiments, with 10 filters at the first layer, and a filter increase factor of 4. All scenarios are trained with the identical network configuration and hyper-parameters. We use standard elastic, intensity, flipping/transposing and rotation augmentations and the Adam optimizer with default parameters (Kingma and Ba, 2014). For more details see Appendix B. To obtain segmentations, we form a topographic map from the outputs of our respective models and employ a seed threshold to locate basins. These are used as seeds in an off-the-shelf watershed transform (Coelho, 2013) and grown until a foreground threshold is reached. See Appendix C for more details on post-processing.

For the nuclei detection task we evaluate the above models with respect to their detection performance, and add a purely detection-based method, namely a model with a single scalar output for regressing Gaussian blobs around center point annotations with sum of squared differences loss (*gauss*, see (Höfener et al., 2018)). The Gaussian regression scenario requires post-processing of the predicted maps to extract center point locations. Non-maximum suppression (nms) is used to locate local maxima above a threshold, with an additional hyper-parameter for the window size.

We use the validation set to determine the number of training epochs and the post-processing hyper-parameters for each scenario individually. For details on the resulting hyper-parameters see Appendix D. All results are averaged over the test set over three independently trained and validated models.

As error metric we use the precision metric used in the kaggle 2018 data science bowl[4]: $AP = \frac{TP}{TP+FP+FN}$. For pixel-wise segmentation of nuclei, we evaluate $AP$ at a range of Intersection over Union (IoU) thresholds. For nuclei center point detection, we define TP, FP and FN as follows: A true positive detection is a center point that lies within a ground truth label and only one such center point counts per label. All other center point

---

4. https://www.kaggle.com/c/data-science-bowl-2018

Table 1: Quantitative evaluation of nuclei segmentation results: Average Precision ($AP = \frac{TP}{TP+FP+FN}$) for multiple intersection over union (IoU) thresholds. STARDIST-3D results from (Weigert et al., 2019). Numbers at all IoU thresholds corroborate the finding that the models trained with our proposed auxiliary task outperform their respective baseline. The best model (3-label +cpv) outperforms the more sophisticated STARDIST-3D model in terms of avAP and in terms of some, but not all, IoU thresholds.

| AP | avAP | $AP_{0.10}$ | $AP_{0.20}$ | $AP_{0.30}$ | $AP_{0.40}$ | $AP_{0.50}$ | $AP_{0.60}$ | $AP_{0.70}$ | $AP_{0.80}$ | $AP_{0.90}$ |
|---|---|---|---|---|---|---|---|---|---|---|
| StarDist-3D | 0.628 | 0.936 | 0.926 | 0.905 | **0.855** | **0.765** | **0.647** | 0.460 | 0.154 | 0.004 |
| sdt | 0.597 | 0.911 | 0.899 | 0.872 | 0.813 | 0.701 | 0.596 | 0.406 | 0.164 | 0.012 |
| sdt + cpv | 0.622 | 0.930 | 0.921 | 0.895 | 0.839 | 0.745 | 0.635 | 0.449 | 0.177 | 0.010 |
| 3-label | 0.629 | 0.928 | 0.919 | 0.892 | 0.833 | 0.734 | 0.623 | 0.461 | **0.231** | **0.041** |
| 3-label + cpv | **0.638** | **0.937** | **0.930** | **0.907** | 0.848 | 0.750 | 0.641 | **0.473** | 0.224 | 0.035 |
| affinities | 0.587 | 0.900 | 0.889 | 0.857 | 0.79 | 0.689 | 0.588 | 0.421 | 0.145 | 0.003 |
| aff. + cpv | 0.608 | 0.921 | 0.912 | 0.886 | 0.826 | 0.726 | 0.615 | 0.430 | 0.154 | 0.006 |

detections count as false positives, and ground truth labels that do not contain any center point detection count as false negatives.

Table 1 and Figure 3 list our quantitative evaluation of segmentation results on the test set. An extended table can be found in Appendix C. Table 2 lists our quantitative evaluation of nuclei detection results. Figure 4 shows exemplary results.

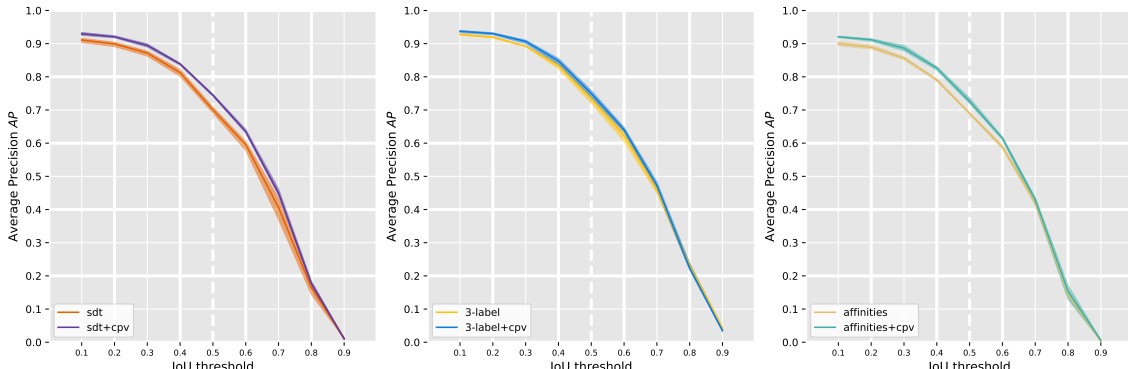

Figure 3: Quantitative evaluation of nuclei segmentation results: Each plot shows the baseline model and the respective model trained with the auxiliary task (+cpv). The shaded area indicates the performance of the respective best and worst performing model of three independent runs. The +cpv models consistently outperform the base models. Best viewed on screen with zoom. See Table 1 for numbers.

Table 2: Top row: Quantitative evaluation of nuclei detection results. All numbers are averaged over three independently trained and validated runs each. The models trained with the auxiliary task (+cpv) consistently perform better than their respective baseline model. The detection performance of the Gaussian regression model is roughly in between the base models and the models with the auxiliary task. Bottom row: For comparison, nuclei segmentation accuracy at an IoU threshold of 0.3, showing that segmentation $AP_{0.3}$ approximately coincides with detection AP.

| metric | sdt | sdt +cpv | 3-label | 3-label +cpv | aff. | aff. +cpv | gauss |
|---|---|---|---|---|---|---|---|
| detection AP | 0.878 | 0.899 | 0.900 | **0.908** | 0.878 | 0.894 | 0.895 |
| segmentation $AP_{0.3}$ | 0.872 | 0.895 | 0.892 | **0.907** | 0.857 | 0.886 | - |

With regards to nuclei segmentation trained on dense ground truth, our proposed auxiliary task boosts segmentation accuracy by up to more than 4% in terms of $AP_{0.5}$. Detection accuracy improves by around 1% – 2% in terms of AP when compared to the respective baselines. These gains are achieved without actually using the predicted vectors to separate clusters of nuclei in the post-processing, just training with the auxiliary task improves the quality of the main prediction. As for nuclei detection trained on sparse ground truth, the Gaussian regression model keeps up surprisingly well and shows better performance than most baseline models and is only slightly outperformed by around 0.5% - 1% by the models trained with the auxiliary task.

## 4. Conclusion

We have proposed an auxiliary task to be used for training deep ConvNets for cell nuclei segmentation in microscopy images. On a database of 28 3d microscopy images, we show that training with this auxiliary task consistently improves performance in terms of nuclei detection and segmentation accuracy. Our proposed auxiliary task is simple and easy to integrate into existing deep learning based 3d segmentation frameworks.

Furthermore, we have shown in a quantitative comparison that nuclei center point detection shows competitive performance when trained on sparse center point ground truth annotations as compared to training on dense ground truth labels.

## Acknowledgments

We thank the authors of (Long et al., 2009) for providing the 3d nuclei data. P.H. and D.K. were funded by the Berlin Institute of Health and the Max-Delbrueck-Center for Molecular Medicine in the Helmholtz Association. P.H. was funded by HFSP grant RGP0021/2018-102. P.H. and D.K. were supported by the HHMI Janelia Visiting Scientist Program.

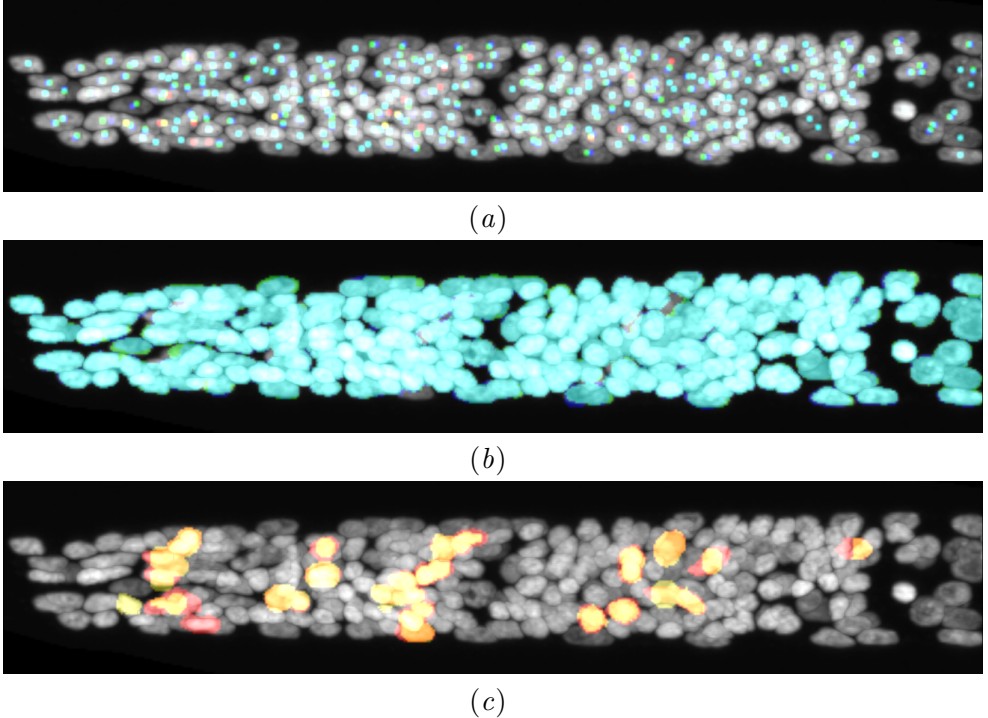

Figure 4: Exemplary results of the head including the nervous system of one worm of the test set using our *3-label+cpv* model. Grayscale: Maximum intensity projection of raw 3d data. True positives: Blue: Ground truth. Green: Prediction. Cyan: Overlay. Others: Red: False positives. Yellow: False negatives. a) Detection results: Most nuclei are detected and localized precisely. b) True positive instances: Only a few pixels at the borders in blue or green, the rest cyan, indicating the match of prediction and ground truth. c) False positive and false negative instances: Most cells are detected, indicated by the high overlap of yellow and red and the missing red and yellow dots in a), however the IoU overlap is too small for some to count as correct segmentations. (see Fig. 5 for whole worms and Fig. 6 for comparison of *3-label model* and *3-label+cpv* model (appendix), visualized with *napari* (napari contributors, 2019), best viewed on screen with zoom and in color)

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

## Appendix A. Visualization

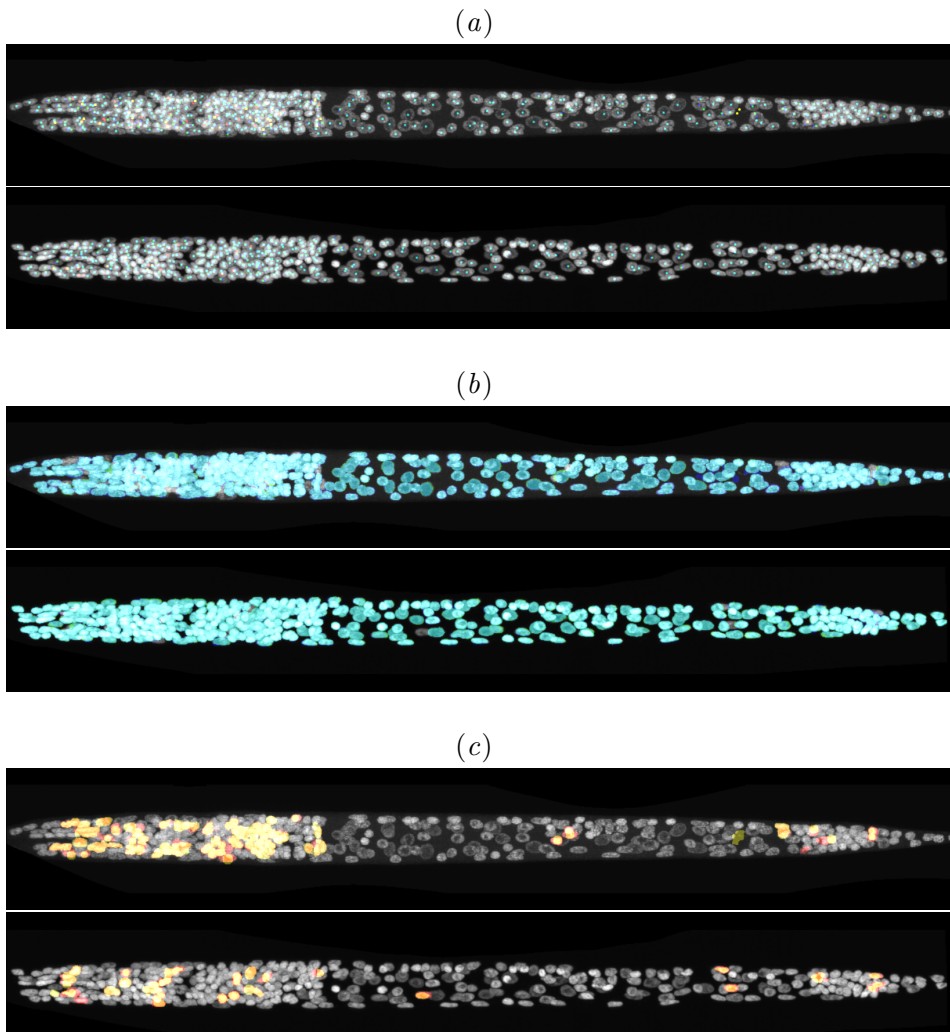

Figure 5: Exemplary results of the worst and best test images using our *3-label+cpv* model. Grayscale: MIP of raw 3d data. True positives: Blue: Ground truth. Green: Prediction. Cyan: Overlay. Others: Red: False positives. Yellow: False negatives. a) Detection results: Most false positive or false negative detections are in the front part of the worm, especially in the densely packed nervous system. b) True positive instances: Only a few pixels at the borders in blue or green, the rest cyan, indicating the match of prediction and ground truth. c) False positive and false negative instances: Most cells are detected, however the IoU overlap (in this case 0.5) is too small for some to count as correct segmentations. (visualized with *napari* (napari contributors, 2019), best viewed on screen with zoom and in color)

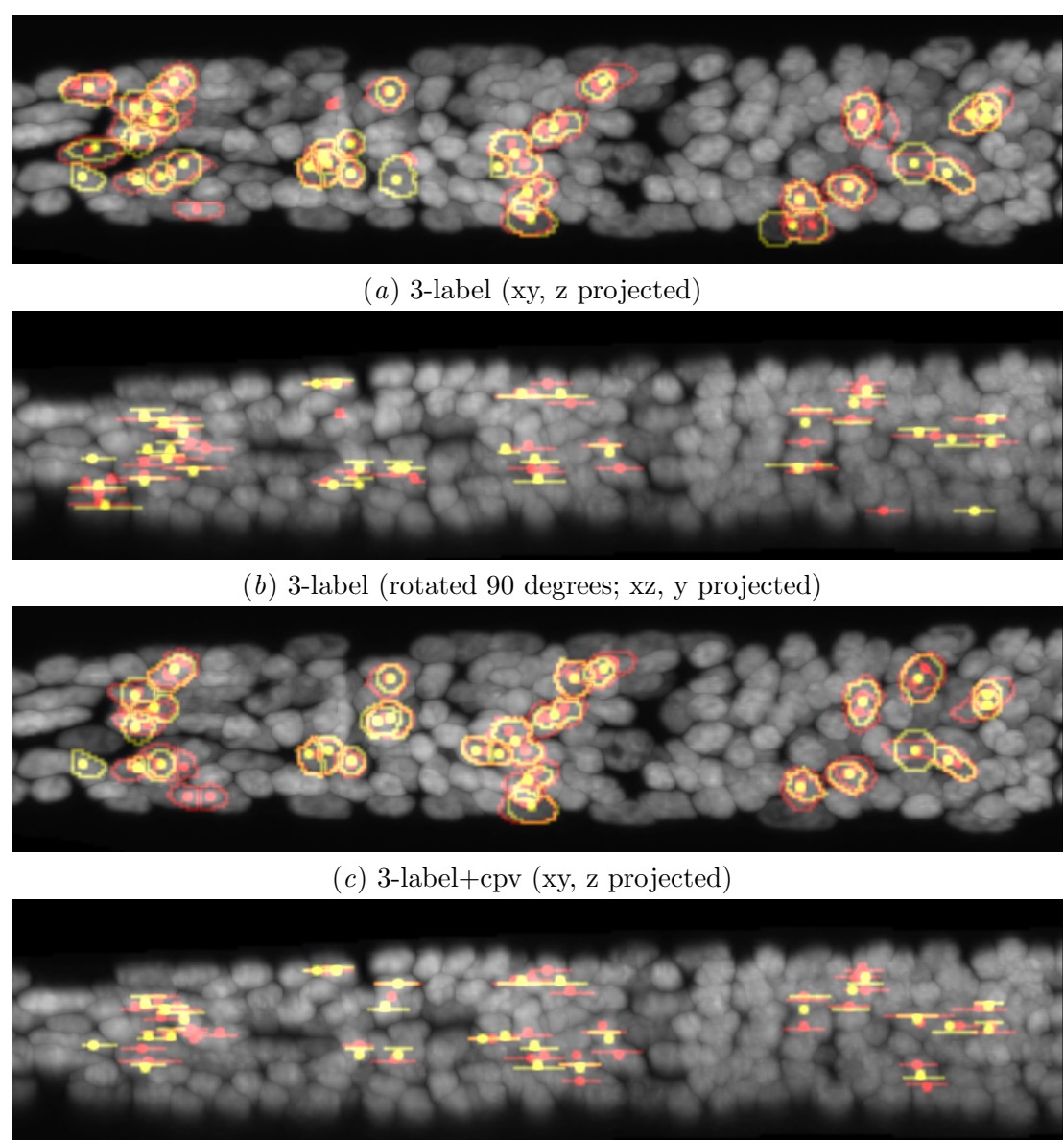

(*a*) 3-label (xy, z projected)

(*b*) 3-label (rotated 90 degrees; xz, y projected)

(*c*) 3-label+cpv (xy, z projected)

(*d*) 3-label+cpv (rotated 90 degrees; xz, y projected)

Figure 6: Exemplary segmentation results obtained with our *3-label* model versus the *3-label+cpv* model. The images show a projection of the 3d data onto a 2d image (not an individual 2d slice). Grayscale: Maximum intensity projection of raw 3d data. Red: False positives (w.r.t. IoU threshold 0.5). Yellow: False negatives (w.r.t. IoU threshold 0.5). There is one dot per FP and FN. The colored lines show the outline of the respective segmentation at the xy-slice of their biggest *xy*-area. Both models make mistakes for a similar subset of nuclei, indicating that those are the ones that are particularly hard to detect and separate. Yet our *3-label+cpv* model has fewer errors. Note, if there is a high overlap of a FP and a FN in both projections, the IoU did not reach the threshold (in this case 0.5) to be counted as a match. (best viewed on screen with zoom and in color)

## Appendix B. Network details

We use the identical 3d U-net architecture for all models. The network has 10 feature maps after the first convolution and has three layers of 2 convolutions each. After each $2\times$ down/upsampling the number of feature maps is in/decreased four-fold. We use $3 \times 3 \times 3$ convolutions, valid padding and ReLU activation functions and constant upsampling. The size of the input window is $230 \times 230 \times 200$, resulting in an output size of $140 \times 140 \times 108$, corresponding nicely to the data sample size of $140 \times 140 \times X$. To train the network we use the Adam optimizer with a fixed learning rate of $1e-4$. We do not explicitly weigh the different loss components (main loss and auxiliary task loss). This is viable as their magnitude is comparable. An exception is the sdt model trained with the auxiliary task. To bring the components to a reasonably similar magnitude we used a factor of 100 for the sdt loss; however, we did not optimize this factor.

The sdt model has a single output and uses the sum of squared differences loss function. The affinity model has four outputs, three for the three direct neighbors (affinities) and one for a separate foreground/background segmentation and uses the binary cross entropy loss with a sigmoid activation per channel. The 3-label model has three outputs, one for the background, one for the boundary and one for the interior and uses the categorical cross entropy loss with a softmax activation. The extended cpv models have each three additional channels for the three dimensional center point vector and use the sum of squared differences loss for the auxiliary task.

The extra computational power required by our auxiliary task is small. GPU-wise, the difference to the base models is minimal, as the networks are identical except for three additional output neurons. The only non-negligible additional computational requirement stems from the use of elastic augmentation during training. For this we need to compute the ground truth center point vectors on-the-fly instead of pre-computing them for all training data. As this is done on the CPU and independent for each training step, it can be done in parallel, and cached for multiple subsequent iterations. We used 20 CPU cores and observed that training time is increased by  10% as compared to baseline training.

## Appendix C. Post-processing details

For the sdt models the network output can directly be used in the watershed transform, all regions smaller than the seed threshold are seeds and are extended until they reach the foreground as determined by the foreground threshold. The euclidean distance transform is computed from the boundary pixel within the instances. Thresholding the foreground at zero leaves the instances slightly small. Validation performance improves by still thresholding at zero but additionally dilating all instances by one[5]. For the 3-label models we use one minus the softmax value of the interior class for the watershed map. One minus the softmax value of the background class is used to define the foreground. For the affinity models we use one minus the average affinity value per pixel for the map. To determine the seeds the affinity values are thresholded individually, if two out of three are above the threshold, the pixel belongs to a seed region. The prediction performance of affinities improves if the boundary between neighboring instances is more than a single pixel. We erode all instances

---

5. https://docs.scipy.org/doc/scipy/reference/generated/scipy.ndimage.binary_dilation.html

before training once to achieve this effect. To reverse the effect of this on the prediction, the resulting instances are dilated once.

We additionally experimented with using the center point vector predictions not only as an auxiliary task during training but also for the post-processing (similar to (Xie et al., 2015)). To this end we transform the three dimensional vector map into a one dimensional height map by accumulating for each pixel a counter of how many vectors point to it. Thresholding this height map results in regions which can be used, as before, as seeds for the watershed. The performance was comparable to using the main prediction. For detection we also list models for which the hyper-parameters were validated and selected wrt. their segmentation performance, instead of their detection performance (*val.* column). (see Table 3 for the extended results)

For the results in Table 1 and Table 2 we used the same training/test/validation data split as (Weigert et al., 2019) to get comparable numbers. To further verify our results we additionally performed 4-fold cross-validation, with the afore mentioned results being one of the folds. The sizes of the training/test/validation sets remain fixed. However, each model of the new folds is only trained once, in contrast to the three independent runs per model for the first fold. The additional numbers confirm our results (see Table 4)

Table 3: Quantitative evaluation results comparing the segmentation results at an IoU threshold of 0.5 and the detection results: We additionally compare seeds computed based on the *main* prediction (depending on the base model) and based on the *cpv* prediction as well as detection performance with the hyper-parameters selected based on the validation performance on the detection task versus on the segmentation task. The models trained with the auxiliary task consistently perform better than their respective baseline model. The detection performance of the Gaussian regression model is better than the baseline models but slightly worse than the models with the auxiliary task (except for the *aff+cpv* model).

| metric | val. | seeds | sdt | sdt +cpv | 3-label | 3-label +cpv | aff. | aff. +cpv | gauss |
|--------|------|-------|-----|----------|---------|--------------|------|-----------|-------|
| segmentation: | | | | | | | | | |
| $AP_{0.5}$ | seg | main | 0.701 | 0.745 | 0.734 | **0.750** | 0.689 | 0.726 | - |
| $AP_{0.5}$ | seg | cpv | - | 0.732 | - | **0.733** | - | 0.725 | - |
| detection: | | | | | | | | | |
| AP | det | main | 0.878 | 0.899 | 0.900 | **0.908** | 0.878 | 0.894 | 0.895 |
| AP | seg | main | 0.879 | 0.898 | 0.896 | 0.909 | 0.871 | 0.892 | - |
| AP | det | cpv | - | 0.889 | - | 0.878 | - | 0.896 | - |
| AP | seg | cpv | - | 0.887 | - | 0.881 | - | 0.890 | - |

Table 4: Quantitative evaluation results for the cross-validation: Overall, the cross-validation confirms the previous results. The models trained with the auxiliary task consistently perform better than their respective baseline model. The Gaussian regression model does slightly better than before and is only outperformed by the *3-label+cpv* model.

| metric | sdt | sdt +cpv | 3-label | 3-label +cpv | aff. | aff. +cpv | gauss |
|---|---|---|---|---|---|---|---|
| segmentation: | | | | | | | |
| avAP | 0.596 | 0.618 | 0.624 | **0.625** | 0.579 | 0.602 | - |
| $AP_{0.5}$ | 0.693 | 0.729 | 0.720 | **0.733** | 0.676 | 0.712 | - |
| $AP_{0.3}$ | 0.870 | 0.896 | 0.886 | **0.897** | 0.855 | 0.885 | - |
| detection: | | | | | | | |
| $AP$ | 0.873 | 0.895 | 0.893 | **0.902** | 0.871 | 0.896 | 0.897 |

## Appendix D. Hyper-parameter details

All models were trained (checkpoints were saved periodically) and validated independently. This leads to different optimal hyper-parameters per model. The best checkpoint and post-processing parameters were determined jointly. In Tables 5, 6, 7 and 8 we list the values for the selected training iteration and post-processing thresholds for all models used to generate the numbers in Table 1 and Table 2 individually. For the models marked as dilated, the resulting instances are dilated once to get the final instances (as described in Appendix C). The *validation* column defines if the hyper-parameters have been selected wrt. validation performance for segmentation or for detection.

Table 5: Overview model hyper-parameters: sdt (+cpv)

| Model | validation on | Iteration | seed threshold | cpv seed threshold | foreground threshold | dilated |
|---|---|---|---|---|---|---|
| sdt (base) | | | | | | |
| I | seg | 60000 | -0.14 | - | 0.0 | yes |
| I | det | 60000 | -0.14 | - | 0.0 | yes |
| II | seg | 100000 | -0.13 | - | 0.0 | yes |
| II | det | 100000 | -0.13 | - | 0.0 | yes |
| III | seg | 100000 | -0.14 | - | 0.0 | yes |
| III | det | 100000 | -0.14 | - | 0.0 | yes |
| sdt +cpv | | | | | | |
| I | seg | 90000 | -0.13 | - | 0.0 | yes |
| I | det | 80000 | -0.13 | - | 0.0 | yes |
| I | seg | 80000 | - | 70 | 0.0 | yes |
| I | det | 80000 | - | 70 | 0.0 | yes |
| II | seg | 90000 | -0.12 | - | 0.0 | yes |
| II | det | 100000 | -0.14 | - | 0.0 | yes |
| II | seg | 90000 | - | 70 | 0.0 | yes |
| II | det | 160000 | - | 60 | 0.0 | yes |
| III | seg | 160000 | -0.12 | - | 0.0 | yes |
| III | det | 200000 | -0.11 | - | 0.0 | yes |
| III | seg | 120000 | - | 70 | 0.0 | yes |
| III | det | 160000 | - | 70 | 0.0 | yes |

Table 6: Overview model hyper-parameters: 3-label (+cpv)

| Model | validation on | Iteration | seed threshold | cpv seed threshold | foreground threshold | dilated |
|---|---|---|---|---|---|---|
| 3-label (base) | | | | | | |
| I | seg | 200000 | 0.7 | - | 0.95 | no |
| I | det | 100000 | 0.8 | - | 0.7 | no |
| II | seg | 100000 | 0.8 | - | 0.95 | no |
| II | det | 100000 | 0.7 | - | 0.95 | no |
| III | seg | 200000 | 0.7 | - | 0.95 | no |
| III | det | 200000 | 0.7 | - | 0.5 | no |
| 3-label +cpv | | | | | | |
| I | seg | 360000 | 0.7 | - | 0.95 | no |
| I | det | 400000 | 0.7 | - | 0.95 | no |
| I | seg | 400000 | - | 80 | 0.9 | no |
| I | det | 400000 | - | 60 | 0.9 | no |
| II | seg | 100000 | 0.7 | - | 0.95 | no |
| II | det | 100000 | 0.7 | - | 0.99 | no |
| II | seg | 200000 | - | 100 | 0.95 | no |
| II | det | 220000 | - | 90 | 0.95 | no |
| III | seg | 220000 | 0.7 | - | 0.95 | no |
| III | det | 220000 | 0.7 | - | 0.5 | no |
| III | seg | 100000 | - | 80 | 0.95 | no |
| III | det | 100000 | - | 70 | 0.95 | no |

Table 7: Overview model hyper-parameters: affinities (+cpv)

| Model | validation on | Iteration | seed threshold | cpv seed threshold | foreground threshold | dilated |
|---|---|---|---|---|---|---|
| affinities (base) | | | | | | |
| I | seg | 300000 | 0.99 | - | 0.99 | yes |
| I | det | 300000 | 0.99 | - | 0.99 | yes |
| II | seg | 200000 | 0.99 | - | 0.99 | yes |
| II | det | 400000 | 0.99 | - | 0.8 | yes |
| III | seg | 300000 | 0.99 | - | 0.99 | yes |
| III | det | 300000 | 0.99 | - | 0.99 | yes |
| affinities +cpv | | | | | | |
| I | seg | 200000 | 0.99 | - | 0.99 | yes |
| I | det | 400000 | 0.99 | - | 0.99 | yes |
| I | seg | 160000 | - | 70 | 0.99 | yes |
| I | det | 300000 | - | 40 | 0.99 | yes |
| II | seg | 200000 | 0.99 | - | 0.99 | yes |
| II | det | 160000 | 0.99 | - | 0.95 | yes |
| II | seg | 100000 | - | 80 | 0.99 | yes |
| II | det | 200000 | - | 50 | 0.99 | yes |
| III | seg | 160000 | 0.99 | - | 0.99 | yes |
| III | det | 160000 | 0.99 | - | 0.8 | yes |
| III | seg | 160000 | - | 80 | 0.99 | yes |
| III | det | 160000 | - | 60 | 0.99 | yes |

Table 8: Overview model hyper-parameters: gauss

| Model | validation on | Iteration | gauss threshold | nms distance | dilated |
|---|---|---|---|---|---|
| gauss | | | | | |
| I | det | 60000 | 0.25 | 3 | no |
| II | det | 80000 | 0.35 | 2 | no |
| III | det | 50000 | 0.25 | 2 | no |

