# OpenReview forum: "An Auxiliary Task for Learning Nuclei Segmentation in 3D Microscopy Images"
_MIDL.io/2020/Conference — MIDL 2020_

### Official Review · AnonReviewer2 · 2020-03-13
**Good approach to improve nucleus segmentation although the core idea is not new**

**Rating:** 3
**Confidence:** 5

**Summary:**

* The authors present a DL method to segment nuclei in 3d microscopy images
* Additionally to learning whether each pixel position belongs to a nucleus, they let the network learn a 3d vector pointing to the center position of the respective nucleus
* They also compare their method with a pure detection-based approach

**Strengths:**

* The paper is well written and easy to follow
* The approach is mainly well described and motivated
* The authors can show improvement (even if very small in some cases) of all methods they extended with their approach

**Weaknesses:**

 Regression of a vector pointing to the center of the nucleus has been done by Xie et al. (2015)*
* Their work should be cited and be differentiated from the authors work
* Adding the auxiliary loss to the main loss requires weighting. In this case, no weighting was described, so we can assume that both losses are just added up without a scaling factor. This *may* lead to a good balancing of the losses, but does not have to. An explanation of that (missing) weighting should be added
* It is not clear to me, whether the vector output of the +cpv methods is used to create the final segmentation. This should be described more clearly

*Xie, Yuanpu, Xiangfei Kong, Fuyong Xing, Fujun Liu, Hai Su, and Lin Yang. “Deep Voting: A Robust Approach Toward Nucleus Localization in Microscopy Images.” In Medical Image Computing and Computer-Assisted Intervention – MICCAI 2015, edited by Nassir Navab, Joachim Hornegger, William M. Wells, and Alejandro F. Frangi, 9351:374–82. Cham: Springer International Publishing, 2015. http://link.springer.com/10.1007/978-3-319-24574-4_45.

**Justification Of Rating:**

* The authors present a well-written paper about their method
* The method has very good aspects such as being easily integratable into other methods
* Results of the method are promising
* Major drawback of the work is that the authors are not relating their own work with Xie et al, who already described vector-based nucleus detection 5 years ago.
* Although this work differs from Xie's method, it should have been mentioned and discussed in the related work section

**Paper Type:**

both

**Questions To Address In The Rebuttal:**

* A short medical motivation to segment nuclei in general and in 3d images in particular would be beneficial
* It is said that the compared "gauss" method is modified (non-maximum suppression), but details are missing. E.g. is the original post-processing step used.

**Special Issue:**

no

---

> ### Author Response · Authors · 2020-03-27
> **reply to MIDL 2020 Conference Paper17 AnonReviewer2**
>
> Thank you for your constructive feedback.
>
> Thank you for pointing us to Xie et al., which indeed we had overlooked. We will add a respective discussion to the paper. We mentioned that the basic idea of regressing center point vectors is not new, yet we only referenced the work of Li et al., 2018. Xie et al. locate points with the highest number of incoming offset vectors (analogously to what we called center point vectors) for nuclei detection. This is similar to an alternative inference approach we describe and evaluate in Appendix C/Table 3, where, like Xie et al., we use the center point vectors during inference to compute the final segmentation results. In contrast, in our proposed method, we use center point vectors only during training. We will clarify this aspect in the paper.
>
> You are correct that no weighting of the two loss components is employed. We briefly mention this but will make it clearer. We wanted to keep the method simple and easily usable. A weight factor would be an additional tunable hyper-parameter. You are also correct that the results might be improved using non-uniform weighting. However, we wanted to show that it works even without this additional tuning.
>
> You ask for a short medical motivation for nuclei segmentation, which we will provide.
>
> You ask about specifics of our gauss model. We identify local maxima above a certain threshold, as in (Hoefener et al., 2018) (though they do not use the term non-maximum suppression). We are not aware of any fundamental differences between our implementation and Hoefener et al., yet cannot guarantee equivalence. We will modify the text to get this across.

---

> > ### Comment · AnonReviewer2 · 2020-03-30
> > **RE: reply to MIDL 2020 Conference Paper17 AnonReviewer2**
> >
> > Regarding the weighting, my point is that not having an explicit weighting still means that there is weighting involved (in this case 1:1). If you do not want to have that that weighting as an additional parameter (which I totally understand and support), you should give a hint why 1:1 weighting is appropriate, e.g. by stating that both parts of the loss have similar magnitude, if that is the case.
> >
> > The rating of the paper is not changed.

---

> > > ### Author Response · Authors · 2020-04-01
> > > **RE: reply to MIDL 2020 Conference Paper17 AnonReviewer2**
> > >
> > > Thank you for clarifying. We will add to the final version that even implicit 1:1 weighting means that there is some weighting involved, and for 1:1 weighting to be appropriate, both parts of the loss need to be of similar magnitude. This is indeed the case, we will state respective numbers for both parts of the loss individually for our 3-label+cpv model.

---

### Official Review · AnonReviewer3 · 2020-03-13
**auxiliary task for segmentation of objects instances that appear in dense clusters**

**Rating:** 3
**Confidence:** 4
**Recommendation:** Poster

**Summary:**

This paper proposes an auxiliary task for segmentation of objects instances that appear in dense clusters.
Target objects are densely packed nuclei captured by a microscope. The proposed auxiliary task is regress vectors that point from each foreground pixel to the center of mass of the respective nucleus.


**Strengths:**

This auxiliary task can be added to architectures of segmentation CNNs.  The experiential results of three segmentation methods with and without the proposed method show the 1-3% improvement of segmentation accuracy.


**Weaknesses:**

The proposed method regress vectors pointing to the center of object and evaluate the regression error for the loss for learning. This idea looks similar to the idea of the auxiliary task with distance transform maps, where the distance from the center to boundary, which is given by ground truth and prediction result, is used for the computation of loss. In the experiments, the authors did not compare the other auxiliary task and loss functions. Theoretical or experimental explanation of the difference among other relevant methods is required for fair validation.


**Justification Of Rating:**

The idea of proposed method is interesting and experimental results showed some improvement of segmentation accuracy by the method. However, theoretical or technical relevance among other methods is not presented. Comparison among the proposed method and other relevant auxiliary tasks are required to show the validity of the proposed method.

**Paper Type:**

methodological development

**Special Issue:**

no

---

> ### Author Response · Authors · 2020-03-27
> **reply to MIDL 2020 Conference Paper17 AnonReviewer3**
>
> Thank you for your thoughtful feedback.
>
> You ask for a comparison to additional models. Do we understand correctly that you would like to see our base models combined with distance transform as an auxiliary task instead of center point vectors? This would certainly be an interesting supplemental experiment, yet we didn't perform it so far due to the following reasoning:
> Our signed distance transform base model (sdt) uses distance transform maps for its main loss. While it is the model that converges the fastest, the precision it yields at test time is inferior to the 3-label model.
> Furthermore, just as the other base models, sdt lacks, compared to our auxiliary task, the property of abruptly changing ground truth values at the boundary between nuclei. Hence, like the other base models, sdt is susceptible to severe errors (e.g. false merges of nuclei) caused by few erroneous predictions. For these reasons, we do not expect that sdt as auxiliary loss would give a considerable boost in accuracy (but as mentioned above we did not test if this is true). Please let us know if you would still like to see this, and/or if you have other specific models in mind which you would like to see compared.

---

### Official Review · AnonReviewer1 · 2020-03-13
**Learning nuclei segmentation via auxiliary task**

**Rating:** 3
**Confidence:** 4
**Recommendation:** Poster

**Summary:**

In this paper, the authors try to segment cells in 3D microscopy images. They use a publicly available dataset to validate their results and compare it to a recent publication on the same task with the same dataset. One of the issue's with segmenting cells in these kinds of images is that clusters cells are often recognized as 1 object. To overcome this issue the authors propose an auxiliary task to point at the center of mass of every cell.

**Strengths:**

- The authors propose a method to ensure that clusters of cells are well-segmented as single objects.
- Comparison to other publications on the same task/dataset.
- Good baseline with multiple approached and recent tasks.
- Clearly described paper, and an approach that could be used

**Weaknesses:**

Some areas for improvement are included below:

* Introduction
- "in terms of average AP." Abbreviation used without explanation.

* Method
- Could the authors give more information about the architecture of the proposed model. This is unclear from the text.

* Experiments
- The method was tested on just 7 images. In my opinion, this isn't a lot, and the authors should consider using cross-validation.
- How much more computational power is needed for this auxiliary task. Because the results only show a minor improvement over the baseline, the method shouldn't be much more computationally expensive.

* Conclusion
- The results of the proposed method are slightly better compared to the baseline models. Could the authors explain how they could improve the results.

**Justification Of Rating:**

The paper proposed a method to better segment clustered objects for cell detection in 3D microscopy images. the method and experimental setup sections leave room for improvement, cause some details are not clear. The results show a slight improvement over a recently published paper but the authors don't explain well how the results could be improved.

**Paper Type:**

both

**Special Issue:**

no

---

> ### Author Response · Authors · 2020-03-27
> **reply to MIDL 2020 Conference Paper17 AnonReviewer1**
>
> Thank you for your constructive feedback.
>
> You're asking "how results could be improved". Do we understand correctly that you would like to see a qualitative assessment that gives an intuition where the quantitative improvements stem from? To this end, we propose to add a figure showing the qualitative difference between the base model and the +cpv model (Figure available here (password: midl2020): https://send.firefox.com/download/df6ee0a7fc13b550/#jY5yWapyIU3HE5qMAV0mQg ). Does this figure help to answer your question in what way the extended +cpv models were able to improve the performance of the base models?
>
> You suggest cross-validation to test on more than 7 images. So far our main priority was to replicate the approach from the StarDist3D paper to get comparable numbers. That said, we fully agree that cross-validation on all 28 images would be a valuable addition in that it will yield a more robust estimate of the improvement obtained by our auxiliary task (even though our current 7 test images do contain ~3600 nuclei). Following your suggestion, we started training additional sets of models for cross-validation on all images. However, unfortunately the new numbers will not come in before the rebuttal deadline due to a limited number of available GPUs.
>
> You ask about network architecture details. Most of the network architecture details are described in Appendix B, but we noticed that we missed mentioning the size of the input window to the network, which we will add. If there are other specific details that you found to be missing, please let us know and we will add those as well.
>
> You ask about extra computational power required by our auxiliary task. GPU-wise, the difference to the baseline models is minimal, as the networks are identical except for three additional output neurons. The only non-negligible additional computational requirement stems from the use of elastic augmentation during training. For this we need to compute the ground truth center point vectors on-the-fly instead of pre-computing them for all training data. As this is done on the CPU and independent for each training step, it can be done in parallel, and cached for multiple subsequent iterations. We used 20 CPU cores and observed that training time is increased by ~10% as compared to baseline training.
>
> Thanks for catching that we missed to explain "AP" in the introduction. We will add a note that average AP is the mean average precision, averaged over a range of IoU thresholds.

---

### Meta-Review · Area_Chair1 · 2020-03-30
**MetaReview of Paper17 by AreaChair1**

**Rating:** 3
**Recommendation For Accepted Papers:** Poster

**Metareview:**

This paper presents a solid approach for nuclei segmentation using auxiliary task learning by introducing a detection problem. While not entirely new from a methodological point of view, reviewers agree that there is value in the specific approach and its validation on the proposed dataset.

**Paper Type:**

both

**Special Issue:**

no

---

### Decision · Program_Chairs · 2020-04-11

Accept